# Optimization of SSVEP-BCI Virtual Reality Stereo Stimulation Parameters Based on Knowledge Graph

**DOI:** 10.3390/brainsci13050710

**Published:** 2023-04-24

**Authors:** Shixuan Zhu, Jingcheng Yang, Peng Ding, Fan Wang, Anmin Gong, Yunfa Fu

**Affiliations:** 1School of Information Engineering and Automation, Kunming University of Science and Technology, Kunming 650032, China; kmustzsx@foxmail.com (S.Z.); yjc1364838637@126.com (J.Y.); auasr_schorr@163.com (P.D.);; 2Brain Cognition and Brain-Computer Intelligence Integration Group, Kunming University of Science and Technology, Kunming 650032, China; 3College of Information Engineering, Engineering University of PAP, Xi’an 710018, China; gonganmincapf@163.com

**Keywords:** SSVEP, VR stereoscopic stimulation, parameter optimization, knowledge graph, BCI

## Abstract

The steady-state visually evoked potential (SSVEP) is an important type of BCI that has various potential applications, including in virtual environments using virtual reality (VR). However, compared to VR research, the majority of visual stimuli used in the SSVEP-BCI are plane stimulation targets (PSTs), with only a few studies using stereo stimulation targets (SSTs). To explore the parameter optimization of the SSVEP-BCI virtual SSTs, this paper presents a parameter knowledge graph. First, an online VR stereoscopic stimulation SSVEP-BCI system is built, and a parameter dictionary for VR stereoscopic stimulation parameters (shape, color, and frequency) is established. The online experimental results of 10 subjects under different parameter combinations were collected, and a knowledge graph was constructed to optimize the SST parameters. The best classification performances of the shape, color, and frequency parameters were sphere (91.85%), blue (94.26%), and 13Hz (95.93%). With various combinations of virtual reality stereo stimulation parameters, the performance of the SSVEP-BCI varies. Using the knowledge graph of the stimulus parameters can help intuitively and effectively select appropriate SST parameters. The knowledge graph of the stereo target stimulation parameters presented in this work is expected to offer a way to convert the application of the SSVEP-BCI and VR.

## 1. Introduction

Brain–computer interfaces (BCIs) comprise a revolutionary human–computer interface that aims to enable direct communication between the brain and external devices by bypassing peripheral nerve and muscle activity, with the goal of improving the quality of life and productivity [1,2,3,4,5,6]. The steady-state-visual-evoked-potential (SSVEP)-based BCI is a significant class of BCI with potential applications [7]. By integrating the SSVEP-BCI with virtual reality (VR), the sense of experience and satisfaction of this technology can be enhanced, promoting its transition into practical applications [8]. Furthermore, VR can expand the use of the SSVEP-BCI from the physical world to VR environments [9,10,11,12,13,14,15]. Therefore, research in this area is of utmost importance.

The performance and user experience of the SSVEP-BCI system are significantly impacted by visual stimulation parameters. VR can provide visual stimulation with more attributes, which is beneficial for relieving the eye discomfort caused by flicker [16,17]. However, the combination of the SSVEP-BCI with VR typically requires the use of head-mounted displays (HMDs), and the system performance is often suboptimal [18,19]. The system performance can be improved by adjusting the stimulation target’s characteristics. Currently, two forms of stimulation targets are used in the SSVEP-BCI-VR. The first is the plane stimulation target (PST), which is induced by two-dimensional stimulation flicker [20,21]. Although this stimulation form is adopted in a three-dimensional scene, it is still a plane stimulus without spatial attributes. The second is the stereoscopic stimulation target (SST), which induces the SSVEP through three-dimensional stimulation flicker. The PST is commonly used as the visual stimulus in the SSVEP-BCI-VR [22,23], while the SST is underutilized. Although the SST parameters (shape, color, and frequency) have a significant impact on the performance of the SSVEP-BCI, they have not been extensively researched.

In this paper, we investigated the online performance of the SSVEP-BCI in VR scenes with different combinations of stereoscopic stimulation parameters (SSTs) including three shapes (cylinders, spheres, and squares), three colors (white, red, and blue), and three frequencies (9 Hz, 11 Hz, and 13 Hz) [24]. The study aimed to construct a parameter dictionary based on the accuracy, variance, frequency difference, and visual fatigue score and to utilize the knowledge graph to clearly characterize the online performance of different SST combinations. The ultimate goal was to customize the SSVEP-BCI with the optimized SST parameters for specific subjects.

## 2. Materials and Methods

### 2.1. Subsection

Ten visually normal students (aged 24 ± 3) were recruited to participate in this study. Prior to the experiment, all participants were provided with a clear explanation of the purpose and procedures of the study and provided written informed consent. The study was conducted in accordance with the Declaration of Helsinki and approved by the Medical Ethics Committee of Kunming University of Science and Technology. All participants approached the experiment with a clear mindset.

### 2.2. On-Line VR Stereo Stimulation SSVEP-BCI System Framework

Figure 1a illustrates the overall structure of the online VR stereo stimulation SSVEP-BCI system, which comprises three main components: EEG signal acquisition, signal processing (preprocessing and classification), and the VR scene. The signal processing module receives the EEG signals via WiFi. The acquired signal is downsampled and filtered using notch and Chebyshev IIR filters and then classified using canonical correlation analysis (CCA). The VR scene responds to the classification results and experimental process control instructions through TCP/IP communication. Figure 1b shows the experimental setup, including the EEG signal acquisition and VR HMD equipment.

### 2.3. Constructing VR Stereo Stimulation Targets and Scenes

Figure 2 shows the VR stereoscopic stimulation targets and scene, which were developed using the Unity 3D platform (Unity Technologies, San Francisco, CA, USA). The size of the target was controlled in the study. In Unity 3D, the establishment of the model follows a unified size regulation, that is 1 unit corresponds to 1 m. In this study, the size of all stimulus targets was unified. The diameter of a sphere, the diameter and height of a cylinder, and the side length of a square were all measured in 1 unit to ensure that they can be considered the same size when viewed by the human eye. To simulate the rotation of the subject’s visual angle in the real world, the display content of the VR-HMD was adjusted based on the rotation of the subject’s head. Based on the parameter set described in the SSVEP-BCI virtual reality SST parameter dictionary in Section 2.6, nine stimulation combinations were created, including white, red, and blue cylinders, spheres, and squares, each with three frequencies (9 Hz, 11 Hz, and 13 Hz). The stimulation parameters were varied by selecting different stimulation combinations.

### 2.4. Experiment

#### 2.4.1. Experimental Process

The experimental field of this study was a quiet, well-lit, spacious room, which can provide the activity space for the subjects. The experimental sessions were scheduled in the morning to ensure that the participants were well-rested. During the experiment, no external human interference was allowed. To achieve optimal wireless signal transmission, the distance between the amplifier and the wireless WiFi was kept within three meters.

First, conductive paste was applied to reduce impedance, and the devices were connected. Next, the stimulation target parameters for the experiment were set on the Unity 3D interface, and the running server waited for the client to connect. Participants were then asked to wear the VR-HMD and adjust the position and pupil spacing. After the adjustments were made, the client sent an instruction to start recording the EEG data. The stimulus target position marker appeared in the VR scene, indicating to the subject to prepare to begin.

The experimental process is illustrated in Figure 3. Each subject was given 10 s of preparation time for each experiment to relax, adjust his/her mindset, and become familiar with the experimental scene. After the preparation period, the target flashed for 4 s, and based on the online classification results, the selected target remained temporarily in the scene for a feedback prompt lasting 1 s. Afterward, the subjects rested for 5 s, and the entire process was repeated for a total of 30 times. After the three targets were selected 10 times, the data were saved after the experiment, and the subjects filled out the questionnaire and switched to the next parameter combination to prepare for the test. A five-minute break with the eyes closed was provided between each round of online parameter experiments, and each subject could perform experiments with up to three parameter combinations per day.

#### 2.4.2. Data Acquisition Process

The electrode layout for the EEG signal acquisition was recorded from three electrodes (O1, Oz, and O${}_{2}$) in the occipital region of the head, and the EEG amplifier adopted a Boruikang (China) NeuSen W wireless digital EEG acquisition amplifier.

#### 2.4.3. Contrast Experiment

We developed a virtual villa scene using VR technology and incorporated a plane flicker target and a stereo flicker target. The parameters of the plane target refer to the VR scene maze paradigm of the Bonkon Koo team [12], and three plane squares were created; each square was equal in size, and the relative interval on the virtual desktop was the same (Figure 4). In contrast, we used a knowledge graph to select the blue square as the optimal parameter to create a stereoscopic stimulus object based on its shape. We conducted three rounds of comparative experiments, and each experiment followed the same process as the formal experiment.

### 2.5. Data Processing Method

#### 2.5.1. Pretreatment

Firstly, the original signal underwent downsampling to achieve a sampling rate of 128 Hz. Then, a 50 Hz notch filter was utilized to eliminate the power frequency interference. Finally, the desired frequency band was obtained through the implementation of a second-order Chebyshev IIR filter with a passband ranging from 5 Hz to 40 Hz.

#### 2.5.2. Classification Algorithm

CCA is a multivariate statistical method employed in data processing to examine the correlation between two sets of variables, revealing their underlying relationship. As a classification algorithm, CCA plays a crucial role in SSVEP classification. Specifically, CCA computes the sin-cos reference signal generated from multi-channel signals and the frequency of the target set stimulation. Subsequently, it selects the stimulation frequency with the highest correlation coefficient as the target outcome of the classification.

Expression of stimulus frequency sin-cos reference signal $Y_{f}$:$$Y_{f}=\left\{\begin{array}{c} \sin\left(2\pi f_{m}t\right)\\ \cos\left(2\pi f_{m}t\right)\\ \vdots \\ \sin\left(2\pi N_{h}f_{m}t\right)\\ \cos2\pi N_{h}f_{m}t \end{array}\right\},t=\frac{1}{F_{s}},\frac{2}{F_{s}},\frac{3}{F_{s}},\cdots ,\frac{N}{F_{s}}$$

In the formula, $N_{h}$ is the number of harmonics of the reference signal (take three in this study), $f_{m}$ is the stimulation frequency, $F_{s}$ is the sampling frequency, and N is the number of samples of the signal. The CCA algorithm calculates the correlation coefficient $\rho_{i}$ between SSVEP signal X and reference template signal $Y_{f}$ through the formula and finds the largest one among all correlation coefficients, and its corresponding frequency is the target frequency.
$$\max_{W_{x},W_{y}}\rho =\frac{E\left[x^{T}y\right]}{\sqrt{E\left[x^{T}x\right]E\left[y^{T}y\right]}}=\frac{E\left[w_{X}^{T}XY^{T}w_{Y}\right]}{\sqrt{E\left[w_{X}^{T}XX^{T}w_{Y}\right]E\left[w_{Y}^{T}YY^{T}w_{Y}\right]}}$$

#### 2.5.3. Performance Index

This paper utilized the accuracy, accuracy variance, frequency deviation, and visual stimulus score as the indicators to evaluate the online performance of various parameter combinations. The statistics for the accuracy included all parameter combinations. The accuracy variance was mainly used to reflect the stability of each parameter combination. The frequency deviation measures the identification accuracy of each frequency parameter under different color shapes. The visual stimulus score indicates the level of visual fatigue caused by different color and shape combinations, with lower scores indicating less visual stimulus and fatigue.

### 2.6. SSVEP-BCI Virtual Reality SST Parameter Dictionary and Knowledge Graph

To optimize the stereoscopic stimulation parameters of SSVEP-BCI virtual reality, this study established a dictionary of stimulation parameters and a knowledge graph of VR-SST. The stimulation parameter dictionary, as illustrated in Figure 5, was categorized into three levels for the ease of retrieval. The first level consisted of nine parameters, including three shapes (cylinder, sphere, cube), three colors (white, red, blue), and three frequencies (9 Hz, 11 Hz, 13 Hz). The second level comprised 18 parameter combinations, formed by adding the color or shape parameter to the selected shape or color in the first level. The third level contained 27 stimulation parameter combinations, with three frequency parameters added to the second level. Each level of stimulus parameters was accompanied by the corresponding interpretations that explained the performance results (classification accuracy, frequency deviation, visual fatigue score, stability) of various parameter combinations, as depicted in the left side of the Figure 5.

Using the stimulus parameter dictionary as a foundation, a VR-SST knowledge graph was developed to categorize the stimulus parameters into three fundamental categories (shape, color, and frequency), corresponding to different attributes of the parameters. As mentioned earlier, there were three parameters for each category. The different combinations of stimulus parameters impacted the SSVEP-BCI performance, with the knowledge graph reflecting the performance differences. By leveraging the three-level structure of the parameter dictionary, the SST can be optimized thrice from the knowledge graph, ultimately yielding the optimal stimulation parameter combination for the SSVEP-BCI.

## 3. Results

### 3.1. Performance Comparison of VR Stereoscopic Stimulation Parameters

#### 3.1.1. Accuracy

We analyzed the experimental data and found that, among the three shapes (Figure 6), the average accuracy was the highest for the sphere (91.85%), the second highest for the square (91.48%), and the lowest for the cylinder (90.18%). From the perspective of color parameters, the comprehensive classification results for blue were the best (94.26%), followed by red (90.19%) and white (89.07%). Finally, from the frequency point of view, the frequency of the 13 Hz target had the highest accuracy (95.93%), and the accuracy of the 9 Hz target had the lowest (85.18%). According to Appendix A, we can see that, among all the parameter combinations, the group with the highest accuracy was the blue sphere at 13 Hz (98.3%), and the group with the worst accuracy was the white sphere at 9 Hz (80%).

The range and variance results of the accuracy for different parameters are shown in the Figure 7, and the most-stable parameters were cylinder, blue, and 13 HZ, respectively.

#### 3.1.2. Frequency Deviation

The recognition accuracy of three frequencies in nine combinations of the shape and color parameters is shown in Figure 8. It can be seen that 13 Hz had less fluctuation than the other two frequencies in the graph, and there was no big fluctuation, while 11 Hz had an obvious fluctuation in the cylindrical parameter experiment. Summing the deviations of each frequency identification, we can obtain the difference in the frequency identification accuracy for the nine parameter sets. In each scintillation experiment, the identification results of the three frequencies were not necessarily completely standard (9 HZ, 11 HZ, and 13 HZ) and sometimes fluctuated up and down. From all the trials with the correct classification, the identification of the three frequencies in each flicker experiment was counted to reflect the accuracy of the frequency identification under different parameters.

#### 3.1.3. Visual Fatigue

By asking subjects to fill out questionnaires after each experiment, the effects of different parameter combinations on the subjects’ visual fatigue were investigated. The visual fatigue questionnaire adopted the Visual Analogy Scale to Evaluate Fatigue Severity (VAS-F) in the Level of Expressed Emotion Scale (LEE). Because some questions on the scale have similar meanings, such as drowsiness and drowsiness, ten differentiated questions were selected. The visual fatigue scores of the experiments with different parameters were as follows: Each item was worth 10 points. The lower the average total score difference, the lower the degree of visual fatigue caused by the corresponding parameters is.

As can be seen from Table 1, the scores of almost all the parameters were higher than the others for drowsiness, tiredness, and wanting to lie down because the induced mode of the SSVEP very easily caused visual fatigue, but in the three combinations of the blue parameters, each score was lower than the other two colors. Similarly, the overall score of the sphere was also lower than the other two shapes (Figure 9).

### 3.2. Knowledge Graph of VR Stereoscopic Stimulation Parameters

In the knowledge graph of the VR stereoscopic stimulation parameters, shape, color, and frequency were represented as abstract nodes, while cylinders, spheres, squares, white, red, blue, 9 Hz, 11 Hz, and 13 Hz were represented as entity nodes. The entity nodes were connected to the abstract nodes via line segments, indicating a relationship between them. The nodes connected by red line segments represent better parameter performance than the other nodes. For instance, the optimal parameter for the shape abstract node was a sphere; the optimal parameter for the sphere node was a blue sphere; the optimal parameter for the blue sphere node was the combination with 11 Hz. Starting from the benchmark node, the primary parameters were selected based on their degree of importance, and the optimal parameter combination can be found to achieve optimal performance, thus optimizing the system performance (Figure 10).

### 3.3. Comparison between SST and PST

The experiment mainly adopted three standards: accuracy, ITR, and subjects’ feelings. The experimental results are shown in Table 2. The results showed that, in the VR scene, using the same experimental process, the stereoscopic stimulus SSVEP with the optimized parameters of the knowledge graph had a better classification accuracy and ITR. In terms of user experience, the overall experience (immersion and desire to use) of the experiment was scored with the standard of a full score after each experiment.

## 4. Discussion

The difference between this proposed study and previous studies is that the research object of this article was stereo stimulus templates in three-dimensional scenes, rather than a plane stimulation target in 2D scenes or a plane stimulation target in three-dimensional scenes. Moreover, the knowledge graph in natural language processing was introduced for parameter optimization, which improved the performance of the 3D-SSVEP system.

Aiming at the SST of the SSVEP-BCI in a VR scene, we built an online VR stereo stimulation SSVEP-BCI system and found that the comprehensive performance of the SST was better than that of the PST. The average accuracy of the sphere stimulation under various color and frequency parameters was 91.85 percent, and the visual fatigue score was 514. The average accuracy of blue stimulation was 94.26 percent under various shape and frequency parameters, and the visual fatigue score was 416. The average 13 Hz stimulation accuracy was 95.93% under various shape and color parameters, and the average test frequency deviation was 0.072 Hz. The accuracy rate reached 98.3 percent when the ideal stimulation parameter combination was blue sphere stimulation at 13 Hz.

According to the statistics of the experimental results, it was found that the online performance of different parameter combinations was indeed different, so it is necessary to select the parameters. From the point of view of shape, among the three shapes, the spherical stimulus parameter was the best, and the visual stimulus scale score of the spherical stimulus parameter was the lowest. Perhaps the spherical SST with fewer edges and corners has a lower degree of visual fatigue. Blue performed the best in terms of color, and it clearly outperformed the other two in terms of accuracy and visual stimulation. Blue was also a better parameter in the parameter research of the plane stimulation SSVEP [25]. White had a lower visual stimulus scale score than red, which indicated that red is more likely to cause visual fatigue. Red light was more likely to make people uncomfortable in the conventional 2D plane stimulation SSVEP, according to prior research [26]. From the frequency point of view, 13 Hz had the best performance and the smallest frequency deviation. Intriguingly, we discovered that the system’s performance in the subsequent villa online comparison experiment stage was not as good as that of the online data collection stage, indicating that the complexity of the scene may affect the performance of the SSVEP. The parameter optimization of the BCI in a VR scene may not be limited to the target object, but the parameters of the whole environment (such as the scene complexity, light intensity, environmental atmosphere, etc.) may also have an impact on the SSVEP.

When counting the experimental results during the knowledge graph construction process, it was discovered that, even with just three colors, three forms, and three frequencies, the burden for the experiments and statistics was still very high. Each subject would be flashed 270 times, and if one wants to seek the performance of a certain parameter combination under a certain premise, one needs to seek the table. It can be predicted that, with the continuous expansion of the subsequent parameter types, the parameter dictionary will be continuously improved, the original look-up table method will become very complicated, and the knowledge graph will become more and more practical with the expansion of the parameter types. The future parameter dictionary will become rather complex with the support of many data, and the benefits of knowledge graphs will continue to be highlighted. The SST proposed in this paper, as shown in Table 2, was superior to the plane target in the accuracy and subject experience after the parameter optimization. Since the SST itself is a part of the VR scene, subjects will not ignore the real-time feedback of the scene itself because of comments. Before the combination with VR, the stimulation target of the SSVEP was basically induced by the plane geometric stimulation, with high accuracy and comfort [27]. It seems to be the best way to combine the plane stimulation directly with VR. For example, in the existing flight simulation [16] and VR maze [19], the PST is directly presented in the subjects’ field of vision, which is quite inconsistent in the VR scene that pursues immersion and fidelity. The subjects may ignore the very important feedback effect of VR itself because they look at the PST next to them.

## 5. Conclusions

The combination of the SSVEP-BCI and VR can promote the conversion and application of this kind of BCI. Based on the online VR stereoscopic stimulation SSVEP-BCI system, this paper established a stimulus parameter dictionary, constructed a knowledge graph, and optimized the stereoscopic stimulation parameters of virtual reality based on the knowledge graph. It was found that the comprehensive performance of the SST was better than that of the PST, and the performance of different combinations of stereoscopic stimulation parameters was different. Using the knowledge graph of the stimulation parameters, the appropriate SST parameters can be selected intuitively and effectively, and the optimal combination of stimulation parameters was the blue sphere at 13 Hz stimulation. It is expected that the knowledge graph of the stereo target stimulation parameters proposed in this study will provide a method for the conversion and application of the SSVEP-BCI and VR.

## Figures and Tables

**Figure 1 brainsci-13-00710-f001:**
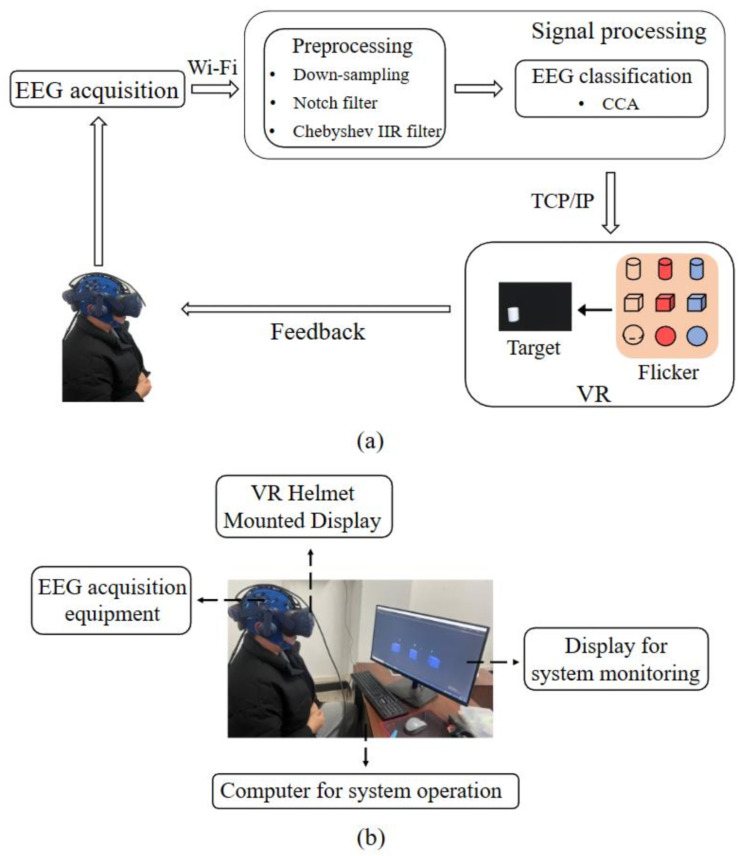
Schematic diagram and experimental setup of the online VR stereo stimulation SSVEP-BCI system. (**a**) Schematic diagram of the system framework. (**b**) Experimental layout.

**Figure 2 brainsci-13-00710-f002:**
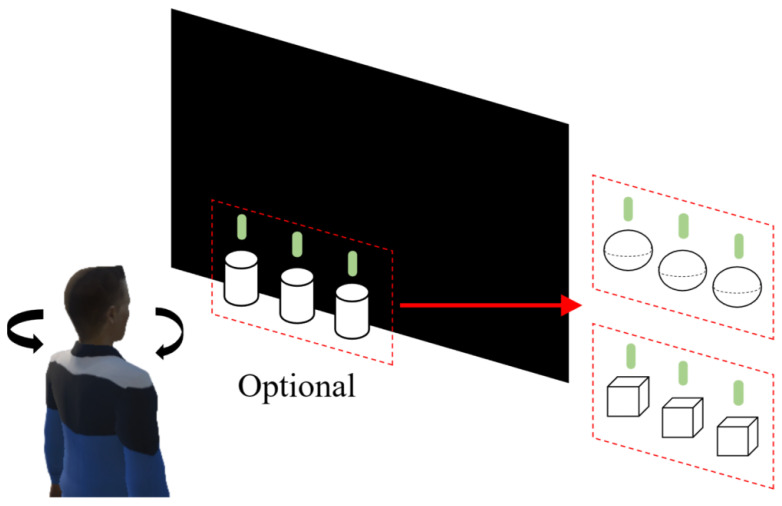
Schematic diagram of the VR scene.

**Figure 3 brainsci-13-00710-f003:**
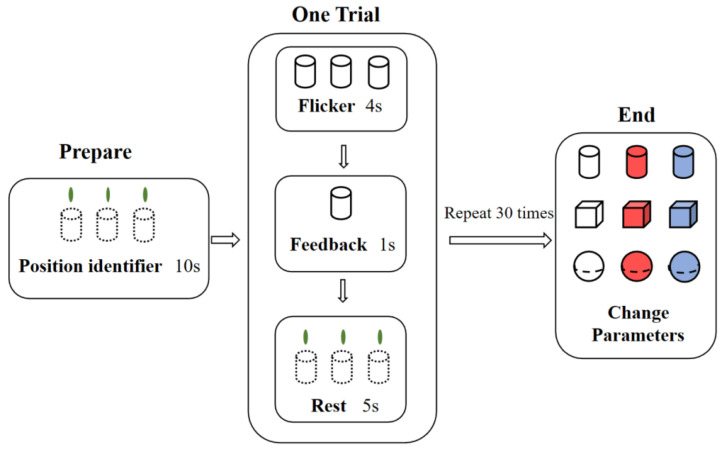
Schematic diagram of the experimental process.

**Figure 4 brainsci-13-00710-f004:**
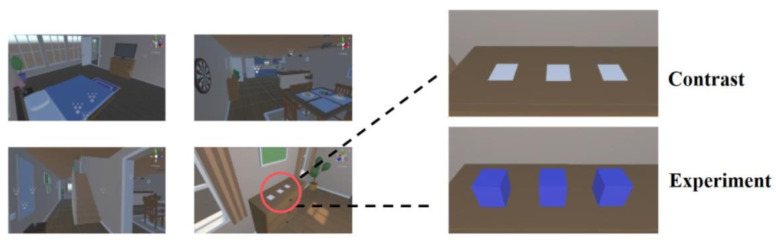
Contrast experimental scene diagram.

**Figure 5 brainsci-13-00710-f005:**
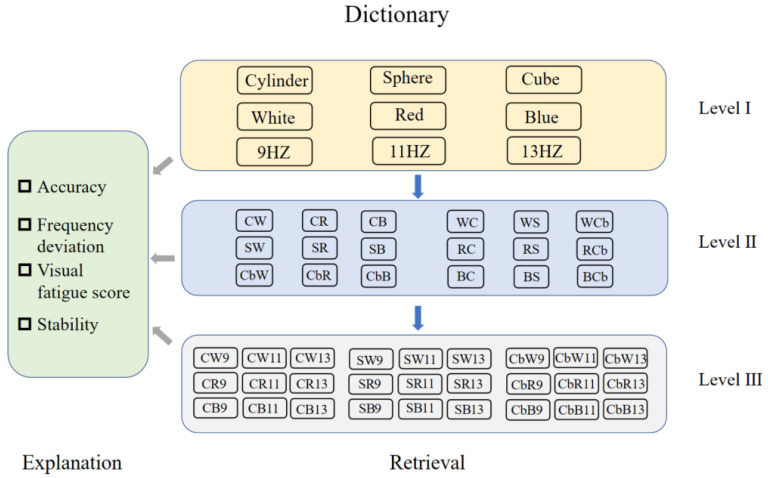
SSVEP-BCI virtual reality SST parameter dictionary.

**Figure 6 brainsci-13-00710-f006:**
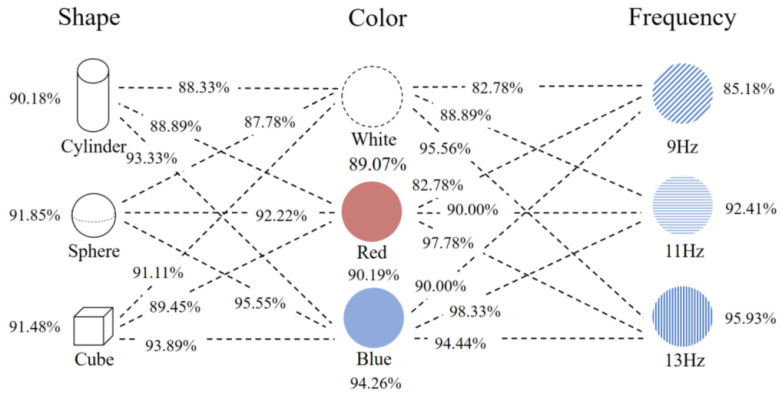
Statistical results of the online experimental accuracy of the virtual stereoscopic stimulation parameters.

**Figure 7 brainsci-13-00710-f007:**
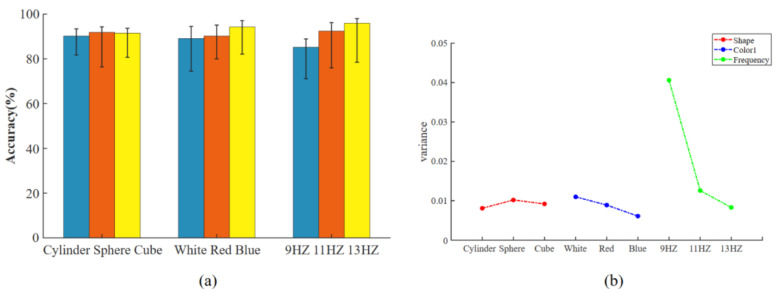
Schematic diagram of the stability of stimulation parameters. (**a**) Schematic diagram of the average accuracy and extreme values. (**b**) Schematic diagram of the stimulus parameter variance.

**Figure 8 brainsci-13-00710-f008:**
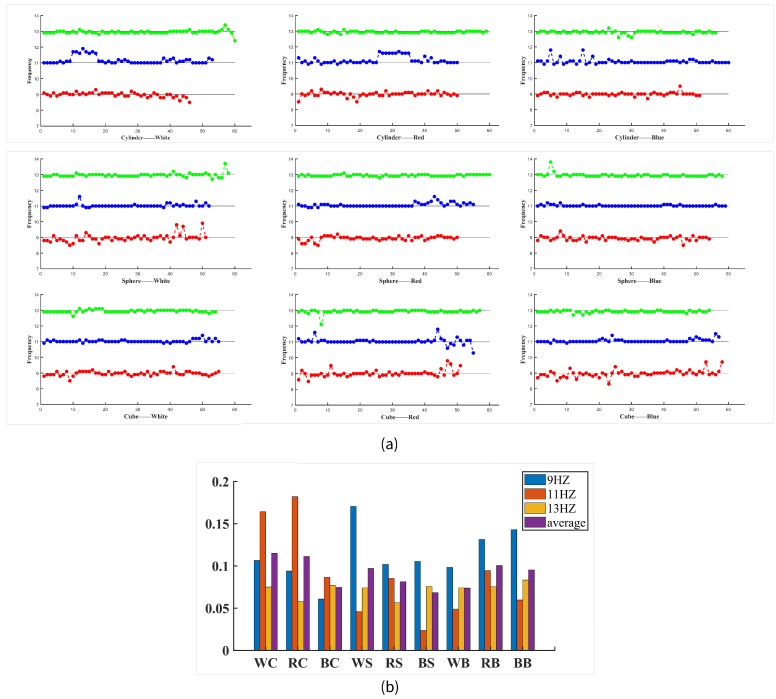
Frequency deviation and fluctuation mean value of different stimulus parameter combinations. (**a**) Frequency deviation fluctuation. (**b**) The average value of the frequency deviation fluctuation was used to represent the stability of the frequency identification of different parameter combinations.

**Figure 9 brainsci-13-00710-f009:**
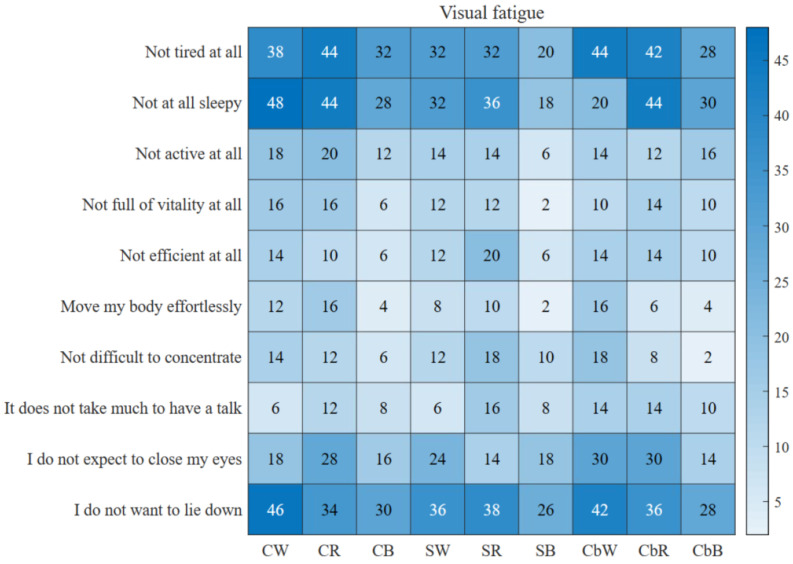
Schematic diagram of visual fatigue degree. The darker the blue color is, the higher the fatigue score is and the lower the fatigue impact caused by the corresponding parameters. The lighter the blue color is, the lower the fatigue score is and the higher the fatigue impact caused by the corresponding parameters is.

**Figure 10 brainsci-13-00710-f010:**
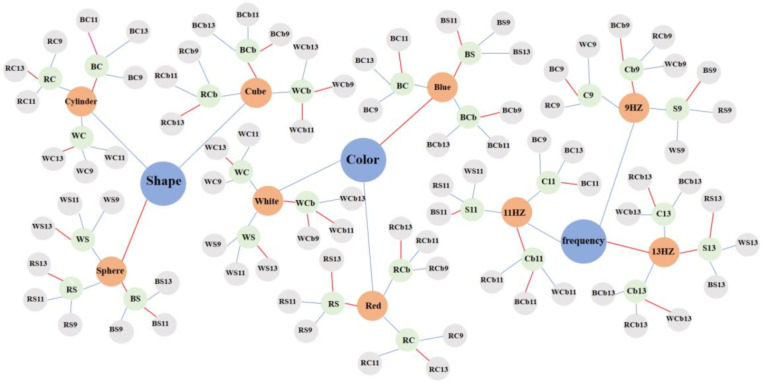
Knowledge graph of SST stimulation parameters.

**Table 1 brainsci-13-00710-t001:** Visual fatigue questionnaire score.

	CW	CR	CB	SW	SR	SB	CbW	CbR	CbB
Not tired at all	38	44	32	32	32	20	44	42	28
Not at all sleepy	48	44	28	32	36	18	20	44	30
Not active at all	18	20	12	14	14	6	14	12	16
Not full of vitality at all	16	16	6	12	12	2	10	14	10
Not efficient at all	14	10	6	12	20	6	14	14	10
Move my body effortlessly	12	16	4	8	10	2	16	6	4
Not difficult to concentrate	14	12	6	12	18	10	18	8	2
It doesn’t take much to have a talk	6	12	8	6	16	8	14	14	10
I don’t expect to close my eyes	18	28	16	24	14	18	30	30	14
I don’t want to lie down	46	34	30	36	38	26	42	36	28
Summary	230	236	148	188	210	116	222	220	152

**Table 2 brainsci-13-00710-t002:** Comparison result of plane target and stereoscopic target.

		Subject 1	Subject 2	Subject 3	Average
Plane target	Accuracy (%)	91.67	75	75	80.55
	ITR (bit/min)	18.65	8.98	8.98	12.20
	Feelings (points)	70	60	60	63.33
Stereoscopic target	Accuracy (%)	81.67	75	90	82.22
	ITR (bit/min)	12.25	8.98	17.42	12.88
	Feelings (points)	80	60	70	70

## Data Availability

The data presented in this study are available on request from the corresponding author. The data are not publicly available due to restrictions eg privacy or ethical.

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
