# Peer review of "Optimization of SSVEP-BCI Virtual Reality Stereo Stimulation Parameters Based on Knowledge Graph"

_brainsci, 2023, doi:10.3390/brainsci13050710_

Round 1

Reviewer 1 Report

Thank you for giving me the opportunity to read the paper by Zhu et al. This paper discusses the potential of using SSVEP-based BCI with virtual reality (VR) and the need for optimizing SSVEP-BCI virtual stereo stimulation targets (SST) parameters. The study presents an online VR stereoscopic stimulation SSVEP-BCI system and a parameter dictionary for VR stereoscopic stimulation parameters. The experimental results of 10 subjects under different parameter combinations were collected and used to construct a knowledge graph to optimize SST parameters. The best classification performances were found for the sphere shape, blue color, and 13Hz frequency parameters.

The paper is well-described and my only concern is the less number of subjects. To that end I would suggest adding a few related papers that used similar number of subjects for similar studies. I would suggest these following papers.

1. Di Flumeri et al. “Brain–Computer Interface-Based Adaptive Automation to Prevent Out-Of-The-Loop Phenomenon in Air Traffic Controllers Dealing With Highly Automated Systems”

2.  Gomez et al. “User Engagement Comparison between Advergames and Traditional Advertising Using EEG: Does the User’s Engagement Influence Purchase Intention?”

3. Perera et al. “Improving EEG-Based Driver Distraction Classification Using Brain Connectivity Estimators”

4. Lakshminarayanan et al. "The effects of subthreshold vibratory noise on cortical activity during motor imagery"

Best Regards

Reviewer 2 Report

Please explain the related study more complete and summarize what the difference between this proposed research and previous study is.

Reviewer 3 Report

In general, the study is interesting in terms of exploring the use of SSVEP and its optimization in a VR setting. I would like to see a discussion on the following points regarding the experiment.

1. Did the authors control the size of used stimuli? It has been shown that the size (number of pixels) of the SSVEP stimulus influences the evoked potential [1]. Hence, it is advisable to account for the effect of the number of pixels used in this study. If the number of pixels is different between the cylinder, square and sphere, how does the size correlates with the final results?

2. The authors used 9, 11 and 13 Hz, which lie in the alpha band, and is believed to contain much noise, thus, should be avoided [2]. Therefore, I would like to see a motivation as to why the authors selected these frequencies? Plus, whether the alpha wave would influence the signal quality in their case or not, especially the “frequency deviation”. Could it be that the alpha wave peaks around 10 Hz made your 9 and 11 Hz responses worse, and since the 13 Hz case is at the boundary of the alpha range is less influenced?

3. Given a population size of 5, is the difference in your result statistically significant? Any statistical test results?

Apart from the presented experimental results, I would also like to read the authors’ opinions and suggestions on the future of VR applications. As in VR games, among other applications, more complicated shapes and colors are typically used than the ones considered in the paper, how would your research results transfer in those cases?  

References

[1] Duszyk, Anna & Bierzyńska, Maria & Radzikowska, Zofia & Milanowski, Piotr & Kuś, Rafał & Suffczynski, Piotr & Michalska, Magdalena & Labęcki, Maciej & Zwoliński, Piotr & Durka, Piotr. (2014). Towards an Optimization of Stimulus Parameters for Brain-Computer Interfaces Based on Steady State Visual Evoked Potentials. PloS one. 9. e112099. 10.1371/journal.pone.0112099.

[2] Norcia AM, Appelbaum LG, Ales JM, Cottereau BR, Rossion B. The steady-state visual evoked potential in vision research: A review. J Vis. 2015;15(6):4. doi: 10.1167/15.6.4. PMID: 26024451; PMCID: PMC4581566.

Round 2

Reviewer 2 Report

Good job.